Development of olfactory epithelium and associated structures in the green iguana, Iguana iguana—light and scanning electron microscopic study

Sapoznikov Olga sapoznikovsco@vfu.cz
Cizek Petr
Tichy Frantisek
Department of Anatomy, Histology and Embryology, University of Veterinary and Pharmaceutical Science Brno , Czech Republic
Esteban María Ángeles
Electronic publication date: 2016 Dec 1
Publication date: 2016
Volume: 4
Electronic Location ID: e2561
Received 2016 May 2; Accepted 2016 Sep 13
Copyright: ©2016 Sapoznikov et al.
Copyright year: 2016
Copyright holder: Sapoznikov et al.
License: This is an open access article distributed under the terms of the Creative Commons Attribution License, which permits unrestricted use, distribution, reproduction and adaptation in any medium and for any purpose provided that it is properly attributed. For attribution, the original author(s), title, publication source (PeerJ) and either DOI or URL of the article must be cited.
License URL: https://creativecommons.org/licenses/by/4.0/

Keywords: Morphology, Sensory epithelium, Green iguana, Nasal cavity, Development

Funding: The authors received no funding for this work.

==============================
The ontogenesis of the nasal cavity has been described in many mammalian species. The situation is different with reptiles, despite the fact that they have become relatively common as pets. In this study we focused on the ontogenesis of the olfactory epithelium, as well as other types of epithelia in the nasal cavity of pre-hatched green iguanas (Iguana iguana). Collection of samples began from day 67 of incubation and continued every four days until hatching. Microscopic examination revealed that significant morphological changes in the nasal cavity began approximately at day 91 of ontogenesis. Approximately at this same stage, the nasal cavity epithelium began to differentiate. The cavity was divided into two compartments by a cartilaginous disc. The ventral compartment bulged rostrally and eventually opened up into the external environment. Three clearly demarcated areas of epithelium in the nasal cavity were visible at day 107.

Introduction

Development of the olfactory structures within the nasal cavity has been described in detail in mammalian species, such as the sheep—Ovis aries (Kocianova et al., 2003), the pig—Sus scrofa (Holubcova, Kocianova & Tichy, 1997), the mouse—Mus musculus (Cuschieri & Bannister, 1975) and even in humans (Kimura et al., 2009). Despite the fact that reptiles have recently become very popular domestic pets, clinical and morphologic researches related to these species are rare. Olfaction itself is an important sense, related to primitive behavior patterns (Suzuki & Osumi, 2015); therefore, the investigation of specific structures ontogenesis in reptiles has become a matter of high importance.

When referring to the ontogenesis of an individual species, no single taxon of reptiles has been widely adopted as a laboratory-amendable developmental model. Valuable morphologic criteria for developmental staging are the development of limbs and pharyngeal arches (Wise, Vickaryous & Russel, 2009). General embryonic development has been described in several reptilian species such as the Calotes versicolor (Muthukkaruppan et al., 1970), the Uta stansburiana (Andrews & Greene, 2011) and the Liolaemus tenuis tenuis (Lemus et al., 1981).

The upper respiratory tract of reptiles includes cavum nasi proprium and conchae. Cavum nasi proprium occupies the area of the head cranial to the eyes (Parsons, 1970). The olfactory region of nasal mucosa in squamate reptiles is restricted to the dorsal aspects of the nasal cavity and nasal conchae and is lined by multi-layered sensory epithelium (Rehorek, Firth & Hutchinson, 2000; Jacobson, 2007). The olfactory region is distinguished from the respiratory region by the presence of submucosal Bowman’s glands and the absence of goblet cells (Rehorek, Firth & Hutchinson, 2000). Some authors, who describe olfactory epithelium in other non-squamate vertebrate species, divide the mature cells of the olfactory epithelium into three main groups—olfactory cells, sustentacular (supporting) cells and basal cells (Kocianova, Tichy & Gorosova, 2001; Morrison & Costanzo, 1990; Kratzing, 1975; Polyzonis et al., 1979). Similar cell populations were also described in reptiles (Kondoh et al., 2012).

Chemosensory olfactory structures of many tetrapods include the olfactory organ and the vomeronasal organ (Rehorek, Firth & Hutchinson, 2000). In reptiles, both main olfactory organ and the vomeronasal organ are derived from the olfactory placode (Suzuki & Osumi, 2015; Rehorek, Firth & Hutchinson, 2000; Holtzman & Halpern, 1990). It is believed that the vomeronasal organ and the olfactory epithelium are homologous structures responsible for the olfactory sense. Neural tracts from both of these structures project to the olfactory bulb of the brain (Parsons, 1970).

The current work presents the development of the olfactory epithelium in the green iguana (Iguana iguana). Development of nasal structures in green iguana has been described by Slaby (1982). The author presented two important developmental stages of the green iguana and mainly concentrated on comparative work between this species and other species of the class Sauropsida. The aim of this study was to investigate the morphology of the developing olfactory epithelium in the green iguana, as well as provide more developmental stages than described by Slaby (1982).

Materials and Methods

The samples were gathered in cooperation with Brno Zoo (the city of Brno, Czech Republic). The incubation of the eggs was done at 27°C and with relative humidity of 90%. The embryos used were all removed from eggs, decapitated and placed into a container with formaldehyde. Sampling of embryos did not require special permissions according to local legislative. Sampling began from day 67 of incubation and repeated every four days until hatching. There were 20 embryos used in total. The last embryo was removed from an egg with incubation period of 135 days.

Light microscopy

The samples were processed by methods of fixation and decalcification which are standard for light microscopy; after the embryos were removed from the eggs, they were immediately fixed in 4% neutral formaldehyde.

Due to the ossification of the skull, most embryonic stages were decalcified in a solution of 5.5% EDTA in 4% formaldehyde. The submersion lasted from three weeks (in early stages) till nine months (in late stages). Decalcification point was determined mechanically.

During dehydration, a graded ethanol series was used: the first bath had concentration of alcohol 30%, the following baths had alcohol concentrations of 50%, 70%, 80%, 96% and 100%. Further on, the samples were immersed into a bath of acetone and three baths of xylene.

After the dehydration process, the samples were embedded in paraffin wax. Serial sections were made transversally (in some early stages) and sagittally, resulting in 4 µm thin slides. The sections were then dried and stained by hematoxylin and eosin.

The samples were investigated by the means of light microscopy and documented by digital photography using an Olympus BX51 light microscope and DP70 digital camera.

Scanning electron microscopy

The samples were dehydrated by alcohol ascending series: 30%, 50%, 70%, 80%, 90%, 96% and 100%. Desiccation was performed by Bal-tec CPD 030 Critical Point Dryer (Bal-Tec, UK) using CO2. The samples were gold-plated by Balzers SCD 040 technique. They were examined and photographed with the TESCAN VEGA TS XM 5136 scanning electron microscope.

Staging of ontogenesis was made according to the criteria set by Wise, Vickaryous & Russel (2009) and marked Wx (x = number of the stage).

Results

Light microscopy

Basal structures of the nasal cavity were visible on day 67 of incubation (stage W36); the cavity was arch-shaped. On day 83 of incubation (W38, Fig. 1), the arch-shaped cavity seemed to have a thin channel which widened dorso-caudally and divided into several projections which were lined by stratified columnar epithelium. In the medial plane, the nasal cavity communicated with the oral cavity by the primary choana. The vomeronasal organ was also visible at this stage. It had the shape of a rounded fissure and was lined by stratified epithelium with hyperchromatic nuclei. The uppermost cellular layer had cilia on its apical surface.

Figure 1 Green iguana.

H & E. Light microscopy. General view of nasal cavity on 83rd day of ontogenesis. Transversal section. Nasal cavity (NC), oral cavity (OC), eye (E) and gland (G) visible.

Significant changes in the shape and size of the nasal cavity begun approximately at day 91 of incubation (W39). From lateral direction, in the middle plane of the nasal cavity, there was a growth of a cartilaginous disc, the future turbinate, which divided the nasal cavity into dorsal and ventral compartments (Fig. 2). Ventral compartment bulged out in rostral direction. It was separated from the dentogingival lamina by a thin bony plate. Laterally, the nasal cavity formed shapes considered to be paranasal sinuses in higher vertebrates. At 99 day of incubation (end of W40), the thin bony plate transformed into the base of the palate. At the 103rd day, a prominent mucosal fold projected into the nasal cavity from lateral direction. In further development, several more mucosal folds projected into the cavity. The folds were more numerous in the area of interconnection with the oral cavity. Opening of the nostrils to outer environment happened at the 111th day (stage W42). From this stage on, there were no significant changes in the shape and the size of the nasal cavity. Medially, the cavity comprised most of the facial part of the skull. Laterally, it narrowed into an arch-shaped slit, which surrounded a wide glandular formation.

Figure 2 Green iguana.

H & E. Light microscopy. Nasal cavity (NC) on the 91st day of ontogenesis. Eye (E) marginally visible.

The lining epithelium of the nasal cavity changed through the stages. At day 91 of incubation (stage W39), the differences among the individual epithelium layers lining the nasal cavity were significantly visible. In the dorsal direction, the epithelium of the nasal cavity was columnar and stratified (Fig. 3). In the ventral plane of the cavity, the height of the epithelium decreased by a sharp line and the lining had three layers (Fig. 4). There were differences also in the characteristics of the cells. While the nasal cavity epithelium was dorsally made of cells with distinct hyperchromatic spherical nuclei and non-voluminous cytoplasm, ventrally there were cells with small nuclei, rich and lightly-stained cytoplasm. The basal layer of those compartments of nasal cavity was composed of cuboidal and columnar cells. Their nuclei were located in the apical part of the cytoplasm. In some locations, the lining of the ventral compartment of nasal cavity was composed from a flat layer of cells. Nevertheless, it was still stratified. Structures resembling cilia in other species were apparent in some places on the surface of the epithelium in the ventral nasal compartment.

Figure 3 Green iguana.

H & E. Light microscopy. Detail of epithelium on the dorsal compartment of nasal cavity (NC). The 91st day of ontogenesis.

Figure 4 Green iguana.

H & E. Light microscopy. Detail of epithelium on the ventral compartment of nasal cavity (NC). The 91st day of ontogenesis.

At day 107 of the incubation (W41), the lining of the nasal cavity could be divided into three areas. Dorsally and rostrally, the mucosa was lined by stratified columnar epithelium with a rough base; its cells were arranged into up to fifteen uniform layers (Fig. 5). Tall but rare cilia could be found on the apical surface of the epithelium. Dorsally and caudally, the epithelium was lower and the cells could be divided into elements with rounded hyperchromatic nuclei found at the lower half of the epithelium and elements with oval bright nuclei located at epithelial surface. Ventrally, the epithelium is composed of uniform and bright cellular population (Fig. 6). This epithelium was lower than the lining of the dorsal surface of the nasal cavity; its cells had columnar shape. On the surface, there were numerous but relatively short cilia. The ventral mucosa created folds that were the most numerous near the transition into the oral cavity.

Figure 5 Green iguana.

H &E. Light microscopy. Detail of dorsal lining epithelium of nasal cavity (NC) on 107th day of ontogenesis.

Figure 6 Green iguana.

H & E. Light microscopy. Detail of ventral lining epithelium of nasal cavity (NC) on 107th day of ontogenesis. Gland (G) is visible.

After the opening of the nostrils (111th day stage—W41), the lining of the initial compartment of the respiratory ways was formed by stratified squamous epithelium which was keratinized in its rostral parts. The structure of the mucosal epithelium remained similar to the previous stages and contained several scattered alveolar glands that drained into the nasal vestibule. The vomeronasal organ was seen at day 111 as a formation enclosed laterally by mucosa. At day 115 of incubation (end of W42), the entire nasal cavity was narrower. In the dorsal part, the mucosal surface appeared wavy due to an unequal height of the ciliated epithelium. Ventrally, mucosal surface was arranged into mucosal folds, which were created by projections of basis of the mucosa. From this stage on, there were no changes within the epithelium, but there was an increased development of the vascularization visible in the connective tissue stroma of the nasal cavity.

At 115 day stage (end of W42), lamina propria was highly vascularized by thin-walled vessels. The vessels gradually became more numerous in the following stages. At the 131 day stage of incubation (end of W42, Fig. 7), thin walled vessels were most numerous in the lamina propria of the rostral part of nasal cavity and its vestibule. This vascularization somewhat resembled Kiesselbach’s plexus. Up to day 135 stage (end of W42), the only significant difference was the expansion of the venous network inside the stroma of mucosal connective tissue. It has been located medially from the glandular formation mentioned  previously.

Figure 7 Green iguana.

H & E. Light microscopy. Nasal cavity (NC) and vestibule (NV) on 131st day of ontogenesis. Oral cavity (OC) lining and gland (G) are visible.

Scanning electron microscopy

At day 87 stage (W39), a cleft which interconnects the nasal and the oral cavity was clearly visible. At the oral part, this cleft connected to an S-shaped pit which protruded into the vomeronasal organ (Fig. 8). At day 103 stage (W41), a mucosal fold appeared on the aboral side of the entrance to the vomeronasal organ (Fig. 9). In the following stage, the fold increased in size and covered almost the entire oral half of the cleft which connected the oral and the nasal cavities. At later stages, the fold developed into a wedge, which protruded caudally and created a fork-like branching of the cleft (Fig. 10).

Figure 8 Balzers SCD 040.

Scanning electron microscopy. Nasal cavity (NC) on the 91st day of ontogenesis. Palate, upper lip (UL) and entrance to vomeronasal organ (VO) visible.

Figure 9 Green iguana.

H & E. Light microscopy. Nasal cavity and associated structures on 103rd day of ontogenesis. Epithelium of nasal cavity (NC), oral cavity (OC) and developing vomeronasal organ (VO) and nasal vestibule (NV) are visible.

Figure 10 Green iguana.

Balzers SCD 040. Scanning electron microscopy. Detail of fold over vomeronasal organ (VO) on 91st day of ontogenesis.

Discussion

Our study presents shaping of the nasal cavity and olfactory epithelium in the embryo of green iguana in the second half of the embryonic development (according to the staging by Wise, Vickaryous & Russel (2009)). Sampling was done at an earlier stage; however, the structures of our interest were not yet visible (which correlates to the staging mentioned above).

It must be mentioned, that no large morphological differences were seen in between each subsequent individual samples (incubation days following one another). Nevertheless, it is important to keep in mind that incubation period (and thus, staging) of embryonic development of iguanas is dependent on some external conditions. Length of incubation cannot be used as a sole criterion for embryonic staging in tetrapods (Licht & Moberly, 1965; Van Damme et al., 1992). For this reason, it is possible to assume that incubation under different or changing conditions would cause a non-continuous development with large morphological gaps.

The shape of the cells which seemed to belong to olfactory epithelium had the appearance of olfactory epithelium in other species. In the mouse, the olfactory epithelium has two types of nuclei from 13th day of gestation on; oval nuclei at apical and lower part of epithelium and rounded at middle cellular layer (Cuschierl & Bannister, 1974). In ovine fetus, oval nuclei are believed to belong to sustentacular cells and spherical nuclei to the sensory cells (Kocianova et al., 2003). Kondoh et al. (2012) described similar cell populations in snakes (Kondoh et al., 2012). There is a difference in intensity of the staining of the nuclei of those two types of cells. In mammals, oval nuclei are darker and rounded or spherical lighter. In the green iguana, the situation is opposite. There are two possible explanations to this fact. The first one could be simply circumstantial, related to the specific moment in fetal development and specific cut and view of the slide. The second explanation is suggested by Rehorek, Firth & Hutchinson (2000) and claims that the staining might have to do with species-specific granules that are present in the sustentacular cells of non-mammals. The description of Kondoh et al. (2012) mentions lipofuscin granules in sensory cells. For this reason, we believe that despite this minor difference when compared to mammals, we deal with the same type of epithelium.

Moreover, the location of the olfactory epithelium is similar to mammalian species (Kumar, Kumar & Singh, 1993; Kumar et al., 2000) and to the description in other reptiles (Jacobson, 2007; Parsons, 1970). This anatomical location has also led us to believe that the dorso-caudal epithelial portion is sensory (olfactory).

Rostrally, the mucosa is lined by pseudostratified epithelium with fifteen layers and a corrugated base. This appearance resembles the regio respiratoria in mammals which is lined by pseudostratified ciliated columnar epithelium with blood vessels present under the surface (Dellmann & Eurell, 1998).

Ventrally, the epithelium is composed from uniform cylindrical and bright cellular population. This epithelium is lower than the lining of the dorsal surface of nasal cavity and has three layers of cells. Cilia are numerous and low. In oral direction, the epithelium of the nasal cavity gradually continues through the choana and changes into the epithelium covering the oral cavity. The epithelium found in the area of the oral cavity and covering the hard palate is squamous with polyhedral shaped cells, which would explain the decrease in size of the lining of the ventral department of nasal cavity. The cilia lead us to believe that this epithelium is also a type of respiratory epithelium, with a somewhat different shape due to its transitional location.

The question is—why does the nasal cavity have those lining differences already during the embryonic development? We believe that the differentiation of the olfactory epithelium is, phylogenically, one of the oldest senses. Moreover, the young must start using their sense of smell immediately after hatching. Thus, smell organs have to be already developed enough before hatching.

The vomeronasal organ of green iguana has similar topographic and structural characteristics to the characteristics present in other tetrapods and mammals (Ciges et al., 1977; Slaby, 1982; Salazar et al., 1996). The roof and the sides of the dome of the vomeronasal organ are lined by sensory epithelium, as mentioned by Kratzing (1975). This organ is derived from olfactory placode, as it does in most terrestrial vertebrates and must be well-lubricated, so odorants can dissolve and trigger the receptors (Rehorek, Firth & Hutchinson, 2000). In our observation, the organ was present already before final shaping of the nasal cavity (from initial stages which we examined), and mucosal folds of the nasal cavity and palate overlapped it, thus creating the vomeronasal channel. Early development might, again, suggest early phylogenic development.

In general, it is challenging to make a direct comparison between the development of the entire nasal cavity in the green iguana and other species, since the length of the gravidity or incubation varies remarkably between different animal species. As mentioned earlier, the ontogenesis in green iguana depends also on some external conditions such as temperature and air humidity. In mammals, those conditions would not be an influencing factor. Nevertheless, when comparing individual structures of the nasal cavity, there is a great deal of resemblance; sensory epithelium and structures appear quite early in embryonic development. The division of nasal epithelium into regions happens before the final shaping of the entire cavity. In the last stages before hatching, the cavity does not develop  remarkably.

Additional Information and Declarations

Competing Interests

Author Contributions

Data Availability

The authors declare there are no competing interests.

Olga Sapoznikov conceived and designed the experiments, performed the experiments, analyzed the data, wrote the paper, prepared figures and/or tables.

Petr Cizek performed the experiments, reviewed drafts of the paper.

Frantisek Tichy contributed reagents/materials/analysis tools, general guiding/advise.

The following information was supplied regarding data availability:

The research in this article did not generate any raw data.

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
