# Peer review of "Development of olfactory epithelium and associated structures in the green iguana, Iguana iguana—light and scanning electron microscopic study"

_PeerJ, doi:10.7717/peerj.2561_

## Round 0.1 · original submission · Major Revisions

Please consider all the suggestions in the revised manuscript.

Please also note that both reviewers provided helpful annotated documents.

Reviewer 1 ·

Basic reporting

English :
English throughout the paper is acceptable, with some corrections to make or phrasing to change (first paragraph of introduction, first paragraph of Materials and Methods, first, second and third paragraphs of Light microscopy, second, sixth and seventh paragraphs of discussion)

Intro & background :
Overall the introduction correct, but the end of the introduction ends abruptly without introducing the following material.
Structure :
Common structure is used and fits the content. Sections are clear.

Figures :
Here is the main issue of this manuscript. Figures are not eligible for publishing as they were presented to us. Compression problems or during writing have made all of the figures (at least the light microscopy ones) useless. Labels and scale bars are not discernable.
The legends would gain from commenting the figures presented.

Raw data :
No high quality pictures were available to download, no additional content.

Experimental design

Scope :
This manuscript fits the scope of PeerJ concerning research and biology/veterinary sciences.

Question/gap :
The data presented here aims to fill a gap concerning reptile ontology.
I suggest additional references:
- DOI: 10.1111/j.1439-0264.2011.01101.x
- PMID: 9929611
- DOI: 10.1111/j.1741-4520.2009.00233.x (Development of olfactory epithelium in the human fetus: scanning electron microscopic observations. Kimura et al. 2009)
- doi: 10.1016/bs.ctdb.2014.11.010 (Neural crest and placode contributions to olfactory development, Suzuki, Osumi, 2015)
- doi: 10.1111/j.1439-0264.2008.00847.x (Scanning electron microscopical study of the lingual epithelium of green iguana (Iguana iguana), Abatte et al. 2008)

Validity of the findings

Results, rationale :
The experiment covers a large part of ontogenesis. One hatched specimen might have given a reference point.
Development descriptions include general anatomy and type of epithelium. A deeper and more specific cell population description would help characterize the tissue (gobelet cells?)
Result description is structured and clear
“We have chosen to believe that” should not be mentioned as such in a scientific paper.
A conclusion on the anatomical location of the olfactory portion of the epithelium is based only on the general analogy to mammal epithelium and lacks observation of proof.

Robust data :
A total number of animals used lacks, showing number of samples.
Despite a bad quality of figures, it seems the samples are well stained and of technical quality.

Conclusion :
The manuscript shows moderation in its conclusion and rejoins the original question.

Additional comments

For the reasons mentioned above, I believe this manuscript deserves revision, in particular its figures, the phrasing of the overall paper, and the precocious comparison regarding mammalian analogy.

Reviewer 2 ·

Basic reporting

- The authors should develop (explain clearly) the originality of this article in comparison with previous studies and the interest of this knowledge, Indeed authors point out that it was described by Slaby in 1982

-It could be interesting to give a quick remaining (in the introduction) on the importance of olfactory system (and the vomeronasal organ) in reptiles/lezards, topography, physiology, vascularisation, innervation etc..

- Figure legend should be self-sufficient, so add more details on the species, coloration etc. For a better understanding, please highlight (arrow, arrow head…) the position of visible structures (vomeronasal organ, primary choana, adjacent structures …)

- Figure 8 was not available in high resolution

- Figures 3 and 4 ; 6 and 7 ; 8 and 9 could be merged to permit easier comparisons

Experimental design

- It might be interesting to explain the reasons for beginning at day 67 and every 4 days. Please explain if samples were made before day 67 and nasal structures were not yet visible.

Validity of the findings

- Observations could be stronger with a statistical justification. How many samples/embryos? What was the statistical analyses to investigate variation between samples and incubation duration ?

Additional comments

The article is well illustrated and valuable. This one could be accepted for publication under minor corrections

---

## Round 0.2 · Minor Revisions

Your manuscript still needs to be improved according to the referee's remaining minor suggestions (also see the annotated PDF from Reviewer 2).

Reviewer 1 ·

Basic reporting

English : The overall english is correct. Line 172, the sentence needs to be edited to make sense.

Intro & background : the overall flow of the introduction and background have been improved and help to grasp the topic

Figures : All the figures are of good quality. Some of the light microscopy could have a whiter background, but it's a detail.

Experimental design

Valuable added information in the number of specimen used.

Validity of the findings

The discussion is much clearer and robust, the comparison to mammals is better suggested.
The paper meets it's descriptive goal.

Reviewer 2 ·

Basic reporting

Figure legends should be self-sufficient so species and coloration (at least) should be added.
Some structures are annotated in legends but not visible on corresponding pictures.

Experimental design

Incubation conditions should be added as development speed is dependant on temperature, hygrometry ... Resuts and conclusions are linked to these conditions

Validity of the findings

No comments

Additional comments

Despite some lacking informations, the article is better at 1st revision step (particularly introduction).

---

## Round 0.3 · accepted · Accept

Congratulations for this work.